# Effects of dietary phosphates from organic and inorganic sources on parameters of phosphorus homeostasis in healthy adult dogs

**Britta Dobenecker** [1] *, **Sven Reese** [2], **Sarah Herbst** [1]

1 Chair of Animal Nutrition and Dietetics, Department of Animal Science, Ludwig-Maximilians- Universität, Munich, Germany, 2 Chair of Anatomy, Histology and Embryology, Department of Animal Science, Ludwig-Maximilians- Universität, Munich, Germany

* dobenecker@lmu.de

**Data Availability Statement:** All relevant data are within the manuscript and its Supporting Information files.

## Abstract

### Background

The impact of dietary phosphorus (P) excess, especially on renal and cardiovascular health, has been investigated in several species, but little is known in dogs.

### Objective

The aim of this study was to examine effects of different P sources on concentration and postprandial kinetics of selected parameters of P homeostasis in dogs.

### Methods

Eight beagles received one control diet (P 0.5% dry matter [DM]) and three high P diets (poultry meal, $NaH_2PO_4$, and $KH_2PO_4$; P 1.7% DM) for 18d. Urine samples were collected pre- and postprandially while faeces were collected quantitatively for 5d and analysed for minerals. On day 18, blood was sampled 1h pre- and 0.5, 1, 1.5, 2, 3, 5 and 7h postprandially.

### Results

Pi ($KH_2PO_4$, $NaH_2PO_4$) but not organic P caused an increased apparent P digestibility and significantly influenced kinetics of serum FGF23, parathyroid hormone, P, CrossLaps and bonespecific alkaline phosphatase, demonstrating a disrupted calcium (Ca) and P homeo-stasis with potential harm for renal, cardiovascular and skeletal health.

### Conclusions

Results of feeding Pi to dogs indicate distinct disturbances of Ca and P metabolism, in con-trast to organic sources. The use of Pi in food can therefore not be considered as safe. Fur-ther research, especially on dose and long-term effects, is warranted.

**Funding:** The authors received no specific funding for this work.

**Competing interests:** The authors have declared that no competing interests exist.

## Introduction

Phosphorus (P) is an essential nutrient for virtually all living organisms and is involved in more than 2000 cellular reactions [1]. Dietary P is present naturally in plant and animal tissue, in the form of organic esters and hydroxyl apatite (organic P). Forms of inorganic P salts (Pi) can also be added to processed food and animal feed to meet P requirements or as additive for technical purposes, including water binding, preservation, texture, colour, and palatability enhancement. This nomenclature has drawbacks, but is established in this scientific field [2]. Furthermore, the use of Pi has increased since the late 1980s [3], when efforts began to reduce the use of sodium compounds as additives during food processing [4]. This trend is also present in pet food, where the recommended daily allowance for P is often exceeded severalfold [5–11]. As these additives are 'Generally Recognized As Safe' (GRAS), labelling is not always mandatory [4,10,12,13]. In humans dietary Pi has been linked to renal, cardiovascular and skeletal diseases [4,10,14–19]. This study was conducted to investigate the impact of dietary Pi on canine P homeostasis and health in general, as little work has been done in this field on dogs.

In humans and animals, a complex endocrine system regulates serum phosphorus (sP) levels via the kidneys, skeleton and gastrointestinal tract. The degree of renal P excretion is the major regulating pathway, with 80–90% of P reabsorbed by the kidneys under normal conditions [20]. Although it is not yet known how the dietary P load is sensed by the body [21], high dietary P intake activates the secretion of fibroblast growth factor 23 (FGF23; [22]). FGF23 is a potent phosphatonin, secreted by osteocytes in case of hyperphosphatemia and high serum $1,25\text{-}(OH)_2\text{-}D_3$ concentrations [23–27]. In rodents, it was demonstrated that FGF23 inhibits $1,25\text{-}(OH)_2\text{-}D_3$ mediated active P uptake from the intestine [25,27,28] and reduces the density of renal NaPi-I, NaPi-IIa and NaPi-IIc cotransporters in the proximal tubules [29,30], increasing P excretion [31]. Little work has been done in veterinary medicine, but a phosphatonin-bone-kidney axis is assumed to also exist in dogs [31,32]. Several breeds were found to reduce their renal excretion of P when fed a diet deficient in P [33]. Parathyroid hormone (PTH) increases renal P excretion and additionally increases serum calcium (sCa; [34]) by decreasing the density of renal P transporters [35] and by stimulating osteoclastic activity [36]. Both hormones FGF23 and PTH act to regain balance of serum Ca and P concentrations. It is assumed that next to other hemostatic measures, dogs seek to maintain a constant serum Ca rather by increasing sP through bone turnover than by adapting intestinal absorption rates [37–41].

In the last several decades, discussion on the adverse effects of dietary Pi additives has intensified, partly because of the increasing prevalence of chronic kidney disease (CKD) in humans [17] and cats [42]. The number of studies reporting detrimental effects of Pi on renal, skeletal and cardiovascular health of humans [17,43], rodents [44,45], cats [46–49] and dogs [50–52] has increased. An impaired ability to handle dietary P load in renal disease [53] causes hyperphosphatemia and the constant stimulation of PTH and FGF23 secretion. This results in secondary renal hyperparathyroidism (SRH) and a loss of bone mass, termed chronic kidney disease-mineral bone disorder (CKD-MBD; [54]). The elevated serum concentrations of sP and FGF23 are associated with cardiovascular disease and increased mortality in human patients with CKD [55–60]. CKD has a lower prevalence in dogs than in cats (0.05–3.74% vs. 7.6%; [61,62]), but SRH and hyperphosphatemia are associated with both feline and canine CKD [63]. Schneider et al. (1980) demonstrated that dogs developed CKD after an excessive intake of Pi (0.8 g $K_2HPO_4$/kg BW; [52]). In dogs, SRH caused by a dietary P excess is accompanied by bone loss and soft tissue calcification, especially in the kidneys and vascular media [64–66]. Harjes et al. (2017) were able to show that FGF23 increases early in CKD cases in dogs, and therefore suggested FGF23 as an early biomarker [67].

In addition to the dietary levels of P, several studies have shown that the source of dietary P is an important factor in adverse effects on i.a. renal health. In cats, the source of P has been found to influence P availability, postprandial serum P concentrations as well as renal excretion [49,68,69]. Pi is not a naturally occurring dietary P source and is significantly more available for absorption from the gastrointestinal tract than organic forms in humans [4]. In animal feed, Pi additives are relatively soluble in water and acidic solutions [9,70], a precondition modulating amount and rate of P absorption. In dogs, Siedler (2018) was able to show effects of P source on several serum parameters including PTH and sP 2 hours (h) postprandially (ppr, [71]). The aim of the present study was to verify the results of Siedler (2018; [71]) and to further investigate the effects of an excessive P intake from different sources on the apparent digestibility (aD) of P and the kinetics of P homeostasis. We hypothesise, that kinetics of biomarkers for bone turnover, as well as FGF23, PTH and sP, will respond to the source and amount of dietary P.

## Animals, materials and methods

This study was conducted between August 2017 and April 2018, using eight healthy beagles (3 ±1 y; 14±1 kg BW, BCS 5/9 according to Laflamme [72]) acquired from the chair's kennel. The animals were housed in outdoor kennels in their familiar groups during day and individually in indoor kennels with concrete floor equipped with pet beds at night. Free access to water was always available. All procedures and protocols were conducted in accordance with the guidelines of the Protection of Animals Act. The study was approved by the representative of the Veterinary Faculty for animal welfare and the Government of Upper Bavaria (reference number AZ 55.2-1-54-2532-30-17). After the study, the two youngest dogs remained in the facility for further research and the other six dogs were rehomed.

Three weeks prior to the start of the study, beagles were fed a complete maintenance food supplemented with casein, gelatin, $CaCO_3$ and lard (when necessary), to ensure a Ca and P intake according to their individual requirements. Each dog was then exposed to four consecutive dietary treatments following the identical order in all dogs. One control treatment (CON), meeting exact P requirements, and three treatments containing high levels of dietary P from an organic or inorganic source (HP; Table 1): 1) poultry carcass meal (HPCM; organic), 2) monosodium phosphate (HPNaP; $NaH_2PO_4$; inorganic), or 3) monopotassium phosphate (HPKP; $KH_2PO_4$; inorganic). The latter ones are both registered as additives (EU: EG 1831/2003 and US: 21 CFR 582) and single feedstuffs (e.g. in the EU: EG 68/2013). Each of the dietary trial periods consisted of 3 d dietary adjustment, 10 d adaptation and 5 d balance period, followed by a washout period of ≥14 d, using the control diet (CON). The CON consisted of a complete and balanced diet (64% tribe, 32% rice and 4% casein; Ca/P = 1.4/1) and served as

**Table 1. Content of crude nutrients, and energy (ME) in the control diet (CON) and 3 high phosphorus diets, containing either poultry carcass meal (HPCM), $NaH_2PO_4$ (HPNaP) or $KH_2PO_4$ (HPKP) as a dietary source of phosphorus.**

| Diet | DM | cP | cFi | cF | ME |
|------|------|------|------|------|------|
| | [%] | [% DM] | | | [MJ/100 g DM] |
| CON | 37 | 37 | 1 | 36 | 2.3 |
| HPCM | 54 | 54 | 0 | 9 | 1.9 |
| HPNaP | 33 | 34 | 4 | 25 | 1.8 |
| HPKP | 34 | 34 | 1 | 25 | 1.8 |

DM = dry matter, cP = crude protein, cF = crude fat, cFi = crude fiber, ME = metabolisable energy.

**Table 2. Content of minerals, total P and P from inorganic sources (Pi), and Ca/P ratio in the control diet (CON) and 3 high phosphorus diets, containing either poultry carcass meal (HPCM), NaH$_2$PO$_4$ (HPNaP) or KH$_2$PO$_4$ (HPKP) as a dietary source of phosphorus.**

| total P | Pi | Ca | Na | K | Mg | Cl | Ca/P |
|---|---|---|---|---|---|---|---|
| | | | [mg/100 g DM] | | | | |
| 595 | 280 | 804 | 162 | 875 | 192 | 981 | 1.4 |
| 2133 | 0 | 4066 | 238 | 658 | 127 | 296 | 1.9 |
| 2571 | 2154 | 3477 | 1700 | 731 | 164 | 674 | 1.4 |
| 2409 | 2079 | 3302 | 151 | 2721 | 58 | 203 | 1.4 |

DM = dry matter.

the basis for two of the HP diets. For the CON diet, the majority of the RDA was met through organic P from dietary components, with CaH$_2$PO$_4$ added to reach the required levels.

In the HP diets, P was supplied at fivefold RDA. To obtain the targeted Ca/P of 1.4/1, CaCO$_3$ was added to CON, HPNaP, HPKP. In HPCM, poultry meal served as a protein source as well as the only P source (57% poultry meal, 43% rice; Ca/P = 1.9/1). Carcass meal has a wide Ca/P ratio [73], and as such the resulting Ca/P was higher in HPCM than in the other diets (1.9/1 vs. 1.4/1; Tables 1 and 2). The feed was cooked and stored at -20˚C. All dogs received the same amount of feed, individually supplemented with lard in case of higher energy requirements of individual animals. To achieve this, the dog with the lowest individual energy requirement served as reference for calculating the basal amount of feed. Minerals and vitamins were supplied using a specifically manufactured supplement to meet or exceed requirements [74]. Feed was apportioned individually to meet each animal's requirements according to their metabolic body weight (BW$^{0.75}$). The feed was offered to the dogs as single meal daily, at the same time and in an identical order. In case of incomplete food consumption, leftover amounts were weighed.

During the balance period, faeces were collected quantitatively and weighed, frozen, lyophilised and pooled per dog, ground and homogenized. Because the keeping of dogs in metabolic cages even for limited periods is not acceptable for the proper authority of the approving process (Government of Upper Bavaria), a quantitative collection of urine was not feasible. Therefore, spontaneous urine was sampled ~2 h pre- and ~2 h postprandially (prpr and ppr, respectively) using a special scoop. After measuring of pH and urine specific gravity (USG;), the samples were stored at -20˚ C. On the last day of the balance trial (day 18), blood was sampled 1 h prpr- and 0.5, 1, 1.5, 2, 3, 5 and 7 h ppr. (V. cephalica antebrachii), always adhering to the same order of dogs in all four trials. Blood samples were allowed to clot for 20 minutes at room temperature before centrifugation (1.260 g) for 10 minutes and serum was aliquoted into several Eppendorf safe-lock cups. For FGF23 measurements, whole blood was allowed to clot for 2 h before centrifugation at 560 g for 15 minutes. Preanalytical artefacts due to haemolytic samples can be ruled out. Feed and faeces samples were analysed for dry matter (DM), metabolisable energy (ME), crude nutrients and major minerals. They were wet digested with 65% HNO$_3$ in a microwave system [75]. Crude nutrients were analysed according to the Weende method (VDLUFA 2012), both in feed and faeces. Apparent digestibility (aD) was calculated. ME was estimated based on gross energy (GE) using analysed GE according to European standards [76]. Serum, food, faeces and urine were analysed for P photometrically using the vanadate molybdate method modified according to Gericke und Kurmies (1952; [77]) at 366 nm. Ca, Na, K were measured by flame photometry, and Mg by atom absorption spectrometry. Urine was analysed for creatinine (Crea) using the Jaffé-method (MicroVue Creatinine Assay Kit, Quidel Corporation, San Diego, USA). As the study design did not allow

collecting urine quantitatively, the values for P were related to Crea (P/Crea) and the difference between the prpr and ppr P/Crea [%] (rel. Diff.$_{prpr}$ P/Crea) was calculated. Serum was stored in Eppendorf safe-lock cups at -20° C for minerals, CrossLaps and bALP. Cups intended for PTH and FGF23 analysis were stored at -80° C. Cups for PTH were sent to ALOMED laboratories (ALOMED, Radolfszell, Germany) on dry ice. A direct luminometric sandwich immunoassay (ILMA) was employed, which uses two polyclonal antibodies against different epitopes of intact human PTH. The first antibody is focused against the N-terminal epitope and acts, through its marking with acridinium-esther, as tracer. The second antibody is focused against the C-terminal epitope and is solid bound. The reference range given by the laboratory for adult dogs is 8–45 pg/ml. CrossLaps are a degradation product of mature collagen type I and therefore a biomarker for bone resorption [78,79]. They were analyzed with an ELISA for human serum samples, validated for dogs (Serum CrossLaps (CTX-I) ELISA kit; ids GmbH, Frankfurt/ Main, Germany). According to literature, reference ranges for dogs are 0.11–1.83 ng/ml [80]. Bone formation was quantified using the marker bone specific alkaline phosphatase (bALP) [33,41,71,78,81]. An enzyme immunoassay validated for dogs (MicroVue Bone BAP EIA; Quidel, San Diego, USA) was used for bALP determination. The reference range for adult dogs is 7.0 ± 2.5 U/L (2–3 years old) and 6.7 ± 3.6 U/L (3–7 years old), respectively [82]. Further marker of bone turnover were not determined due to limited amounts of serum available for all laboratory analyses. To determine values for FGF23, a sandwich ELISA with an anti-human FGF23 mouse monoclonal antibody was used (KAINOS Laboratories Inc., Tokyo, Japan). The kit is validated for dogs [67] and used routinely in canine trials [67,83,84]. To our knowledge, no reference range has been defined for dogs, but Harjes et al. (2017) reported of values between 211–449 (mean 315) pg/ml in healthy dogs (n = 10; [67]).

Statistical analysis was performed by using SPSS 24.0 (IBM). Every dog underwent the control phase and the three trials, therefore data were analysed using a repeated measures design. Data were visually checked for normality using q-q plots. Where data were normally distributed, parametric methods were applied; means ± standard deviation are presented. A one-way ANOVA for repeated measurements was conducted to compare the values of each dog across the control and three treatments. For pairwise comparisons, a post hoc test with Bonferroni correction was used. To evaluate the degree of association between aD P and $AUC_{0-7}$ sP (not normally distributed), the non-parametric correlation coefficient Spearman's rho was calculated. A p-value <0.05 was considered significant, ≤0.001 highly significant.

## Results

All dogs remained clinically healthy during this study, as assessed by daily visual inspection and detailed weekly health checks by the keepers. The meals were completely consumed. The mean DM intake amounted to 11±1, 14±0, 13±1 and 16±2 g/kg BW/d in diets CON, HPCM, HPNaP and HPKP, respectively. Appetite was slightly reduced in two dogs during trial HPKP, resulting in a prolonged period of food intake during the balance trial (maximum 1 h). All dogs maintained their bodyweight with a mean standard deviation of 0.7 kg and a body condition score (BCS) of 5/9 [72]. No abnormalities were detected in faeces quality during all four trials.

The total P intake was similar in all 3 HP diets, whereas the amount of Pi intake was highest in diet HPNaP and HPKP, as intended (Table 3). There was a significant effect of P source on the intake of Na and K, with the highest Na intake recorded in the HPNaP diet and the highest K intake in the HPKP diet. The Mg intake in all 3 HP diets was higher than in the CON diet, but more distinctly increased in diets HPCM and HPNaP.

**Table 3. Absolute amount of major mineral intake [mg/kg BW$^{0.75}$/d] in dogs fed a control (CON) and 3 high phosphorus diets, containing either poultry carcass meal (HPCM), NaH$_2$PO$_4$ (HPNaP) or KH$_2$PO$_4$ (HPKP) as a P source.**

| Mineral intake | P | Pi | Ca | Na | K |
|---|---|---|---|---|---|
| | [mg/kg BW$^{0.75}$/d] | | | | |
| CON | 97 ± 3 [a] | 44 ± 1 [a] | 131 ± 4 [a] | 26 ± 1 [a] | 136 ± 3 [a] |
| HPCM | 492 ± 11 [b] | 0 ± 0 [b] | 929 ± 16 [b] | 55 ± 1 [b] | 150 ± 4 [b] |
| HPNaP | 466 ± 7 [a,b] | 396 ± 7 [c] | 636 ± 7 [a,b] | 312 ± 6 [c] | 134 ± 2 [a] |
| HPKP | 493 ± 9 [b] | 415 ± 7 [d] | 665 ± 5 [b] | 31 ± 1 [d] | 557 ± 9 [c] |

Columns not sharing a superscript letter are significantly different (p<0.05).

The aD of P was similar in the CON diet and in both diets with added Pi sources, while the organic P source led to a significantly lower value (22±5 vs. CON 54±9, HPNaP 53±9, HPKP 48±5; p<0.05). The sP concentration was also influenced significantly by the P source (Fig 1).

The HPCM diet resulted in an almost identical AUC$_{0-7}$ sP as the CON diet. In contrast, both Pi additions led to a significantly higher AUC$_{0-7}$ sP (Fig 1A). There was only a weak correlation between aD of P and AUC$_{0-7}$ sP for P excess from organic and inorganic sources (organic P: Spearman's rho -0.074, p = 0.73; Pi: Spearman's rho 0.257, p = 0.109). The ppr. sP concentrations showed no marked increase after feeding diets CON and HPCM, with the highest values determined 7 h ppr. (Fig 1B). The two Pi sources (HPKP, HPNaP) caused a prolonged sP concentration that was clearly above reference range, with peak values after 3 h. In diets CON, HPCM and HPKP, sCa stayed within the reference range at all times (S1 Table), and the AUC$_{0-7}$ (sCa) was similar (S2 Table). Feeding diet HPNaP caused a lower AUC$_{0-7}$ (sCa), a difference statistically significant when compared to diet HPCM (p = 0.026). The consequence of the distinct increase of sP was an elevated AUC$_{0-7}$ for the serum Ca x P product (sCaP) in both diets with Pi addition (HPKP and HPNaP; p≤0.001; Fig 2).

The peak of sCaP was at 3 h ppr, while the recommended threshold of 55 mg$^2$/dl$^2$ [55] was already exceeded from 1 h ppr onwards (S3 Table). HPCM showed no significant difference from CON (p = 0.3) for the AUC$_{0-7}$ sCaP in all high P diets. FGF23 differed significantly between the diets. The Pi sources led to the highest serum concentrations, still significantly elevated in the fasted animals, and differing significantly from the diets CON and HPCM (HPKP:

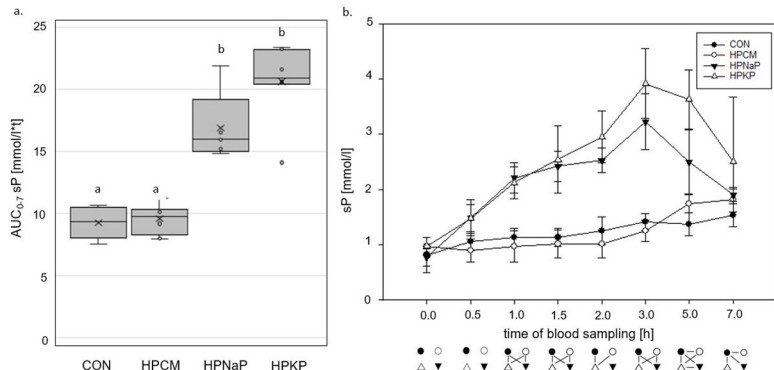

**Fig 1. Serum P (sP) concentrations. a.** Area under the curve from 0 until 7 hours postprandially (AUC$_{0-7}$) of sP [mmol/l*t] in adult dogs fed a control diet and 3 different high P diets for 18 days. Boxes not sharing a superscript letter are significantly different (p<0.05) **b.** Kinetics from 0 to 7 hours postprandially of sP [mmol/l] in adult dogs fed a control and 3 different high P diets. Connected symbols below the graph are significantly different (p<0.05) at the corresponding time point. Reference range for healthy adult dogs: 0.7–1.6 mmol/l [85].

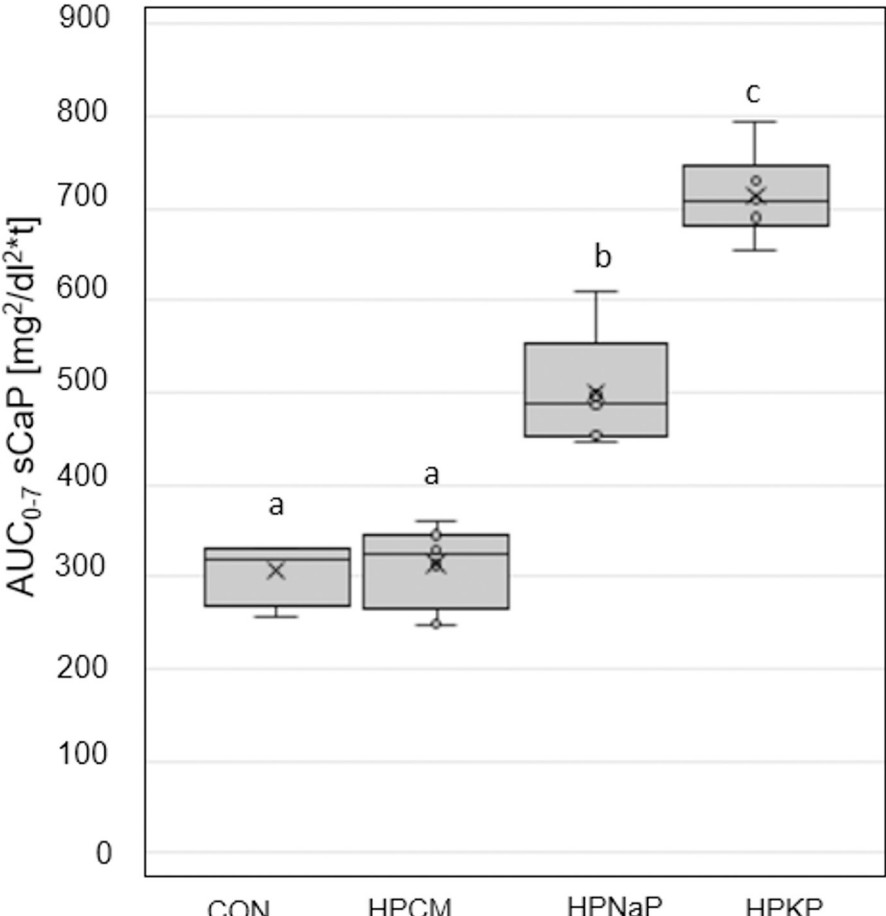

**Fig 2. $AUC_{0-7}$ of sCaP [$mg^2/dl^{2*}t$] in adult dogs fed a control and 3 high P diets for 18 days.** Boxes not sharing a superscript letter are significantly different ($p<0.05$).

$p<0.001$; HPNaP: $<0.05$). Ppr the concentrations slightly decreased over time (Fig 3B). Adding $KH_2PO_4$ to the diet led to a significant higher $AUC_{0-7}$ FGF23 than in all other diets (Fig 3A).

Prpr PTH concentrations were similar in all 4 groups. Ppr they differed significantly between CON and HPCM on one hand and Pi sources (HPNaP, HPKP) on the other with values clearly above reference range (Fig 4B). Both Pi sources led to a rapid increase of serum PTH concentrations and a peak 3 h ppr. CON and HPCM diets did not influence PTH concentrations (Fig 4).

In a similar manner, the CON and HPCM diets had no significant effect on serum concentrations of the catabolic bone marker CrossLaps (S4 Table). Kinetics of CrossLaps correlate with sP kinetics ($R^2 = 0.32$). Diets HPNaP and HPKP led to a severe increase of serum CrossLaps with peak values 3 h ppr and the highest values recorded for the HPKP. The $AUC_{0-7}$ CrossLaps of both differed significantly from the CON and HPCM treatments.

The bone formation marker bALP was within the reference range at all times during the CON and HPCM treatments [82]. However, Pi feeding from both $NaH_2PO_4$ and $KH_2PO_4$ caused a distinct increase of bALP above the reference range (S5 Table).

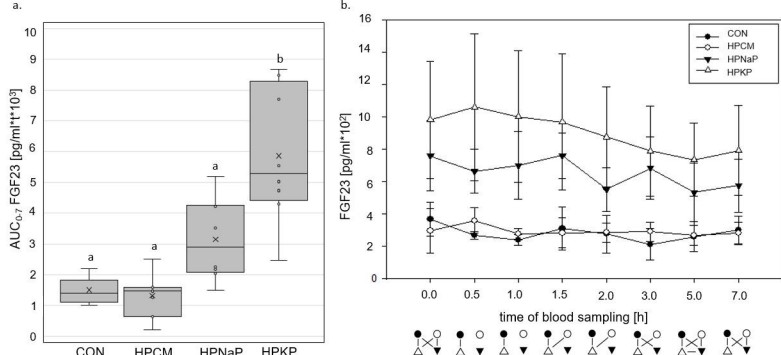

**Fig 3. Serum FGF23 concentrations. a**. Area under the curve from 0 until 7 hours postprandially ($AUC_{0-7}$) of FGF23 [$pg/ml^* t^* 10^3$] in adult dogs fed a control and 3 different high P diets for 18 days. Boxes not sharing a superscript letter are significantly different ($p<0.05$) **b**. Kinetics from 0 to 7 hours postprandially of FGF23 [pg/ml] in adult dogs fed a control and 3 different high P diets. Connected symbols below the graph are significantly different ($p<0.05$) at the corresponding time point. Values for healthy dogs, according to literature: 315 (211–449) pg/ml [67].

In the urine, the rel. $Diff._{prpr}$ P/Crea [%] ~2 h prpr and ppr revealed a significant increase in both Pi treatments, with highest values in HPNaP (389 ± 250%; HPKP 205 ± 120%). CON and HPCM diets led to significantly decreased ppr P/Crea ratios.

## Discussion

Modern pet food often contains significantly higher levels of P than the recommended daily allowance in animals, with levels sometimes exceeding the requirements more than fivefold [5–11,86]. Moreover, not only the amount of total P but also of highly soluble P ($P_{sol1}$) can be extremely high in processed feed [9]. The aim of this study was to investigate the effect of high P diets in practically relevant amounts, containing either organic or inorganic P on healthy dogs. In support of Siedler (2018; [71]), we found that the two highly soluble Pi sources ($NaH_2PO_4$; $KH_2PO_4$), in contrast to the poultry meal (as an organic P source), caused distinct effects on Ca and P homeostasis. We also explored the kinetics of dietary high P on parameters of P and Ca homeostasis and found significantly increased serum concentrations of FGF23, PTH and other parameters after feeding Pi sources.

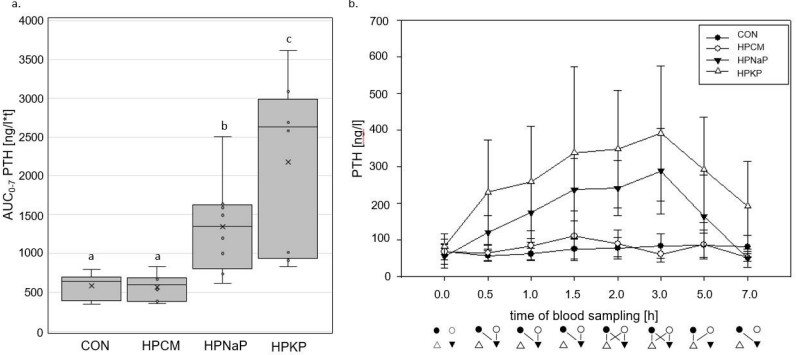

**Fig 4. Serum PTH concentrations. a.** $AUC_{0-7}$ of PTH [$ng/l^*t$] in adult dogs fed a control and 3 different high P diets for 18 days. Boxes not sharing a superscript letter are significantly different ($p<0.05$) **b**. Kinetics from 0 to 7 hours postprandially of PTH [ng/l] in adult dogs fed a control and 3 different high phosphorus diets for 18 days. Connected symbols below the graph are significantly different ($p<0.05$) at the corresponding time point. Reference range for healthy adult dogs: 8–45 ng/l (ALOMED laboratories).

Siedler (2018; [71]) took blood samples during fasting and 2 h after a single daily meal to measure selected parameters of P and Ca homeostasis in dogs. However, it is recommended to sample blood multiple times over a period of 8 h for adequate evaluation of the P status in humans [87]. Jowsey et al. (1974) sampled their trial dogs prpr and 0.5, 1, 2, 4, 6 and 7.5 h after feeding a diet with Pi excess to measure PTH and sP [51]. We therefore chose to take blood samples at 1 h prpr- and 0.5, 1, 1.5, 2, 3, 5 and 7 h ppr, which gave a comprehensive view of the effect of each meal. To our knowledge, apart from Jowsey et al. (1974), no kinetic studies on the homeostasis of orally administered P in dogs are available in literature. In addition to serum P and PTH, bALP and CL [51,71], we also analyzed FGF23 and further major minerals (Ca, Na, K, Mg). P can only be absorbed if it is soluble [88], and as Pi from $KH_2PO_4$ and $NaH_2PO_4$ is highly water soluble [70,89], this inorganic source may therefore be more readily available. Next to the dietary P source [37,90,91], the digestibility of P is also dependent on other elements, including Ca [37,40,92], and on the pH of chyme [93,94]. The organic P dietary treatment (HPCM) produced a significantly lower aD of P than the other diets. Possible explanations for this lower aD of P are the low solubility of P from meat and bone meal (only 4% after 1 minute in water vs. 33% in HPKP and 56% in HPNaP, method of Lineva et al., 2017 [89]), the higher Ca/P ratio [37,39], and the lower aD of DM (p<0.05). It has been shown that high dietary Ca concentrations may decrease P digestibility in dogs [37,95] and cats [96,97], probably by the formation of insoluble Ca/P complexes [39,40,98], most probably $Ca_3(PO4)_2$. However, a recent study disproved this to be the case when Pi sources are concerned: In contrast to organic P sources, here only a weak correlation existed between faecal Ca and faecal P excretion [99]. It was hypothesised, that absorption of highly soluble P salts happens early in the gastrointestinal tract of dogs mostly via passive mechanisms, before Ca-P-complex formation can reduce availability. Furthermore, it was shown in dogs that an increase of the Ca/P from 1.4/1 to 1.9/1 had no effect on aD of P or prevented a significant increase of sP and sCaP in a HP diet with a highly soluble Pi source [100]. This implies that a mere increase of the Ca/P ratio in a product with considerable amounts of soluble Pi salts does not suffice to protect the user from a high P burden. Still, more data are needed to further strengthen the hypothesis that the Ca/P ratio affects the P availability to a relevant degree only when organic P salts are used. The significantly lower aD of DM in the diet HPCM will have affected the aD of P too, as the faecal excretion of P and other minerals is influenced by faecal DM excretion [101].

Furthermore, aD of P did not correlate well enough with $AUC_{0-7}$ sP ($R^2 = 0.3$; Fig 5) to consider the aD as an appropriate tool to predict sP. An aD P of about 30% caused an $AUC_{0-7}$ sP between 7 and 10 mmol/l*t in organic P sources and an $AUC_{0-7}$ sP of up to 22 mmol/l*t in Pi sources; a biologically highly relevant difference.

Our results also showed that Pi sources form a separate cluster from HPCM with higher $AUC_{0-7}$ sP. In a novel approach, we also determined the markers of P homeostasis over 7 h ppr to follow their kinetics. Therefore, it was possible to demonstrate that even at 1h ppr sP was significantly increased in Pi diets, with peak levels at 3 h ppr. In contrast, CON and HPCM sP increased only slightly. This corresponds well with data from Jowsey et al. (1974) with similar sP concentrations postprandially, as well as Harris et al. (1976; [51,102]). Increased sP concentrations, especially over an extended period, should be avoided [103–105]. In humans, it is well known that even sP concentrations reaching the upper limit of normal range are associated with a higher hazard ratio for death [103,104,106]. Osuka and Razzaque (2012) postulated that a P toxicity appears even in normophosphatemia. They conclude that reducing the P burden and maintaining P balance through adequate dietary intake is crucial for normal health, as P toxicity can cause irreversible organ damage [107]. Kestenbaum et al. (2005) identified a significantly increased risk for death when sP levels were above 3.5 mg/dl (= 1.13 mmol/l; normal range: 1.12–1.45 mmol/l; [108]) in a cohort study of more than 6000

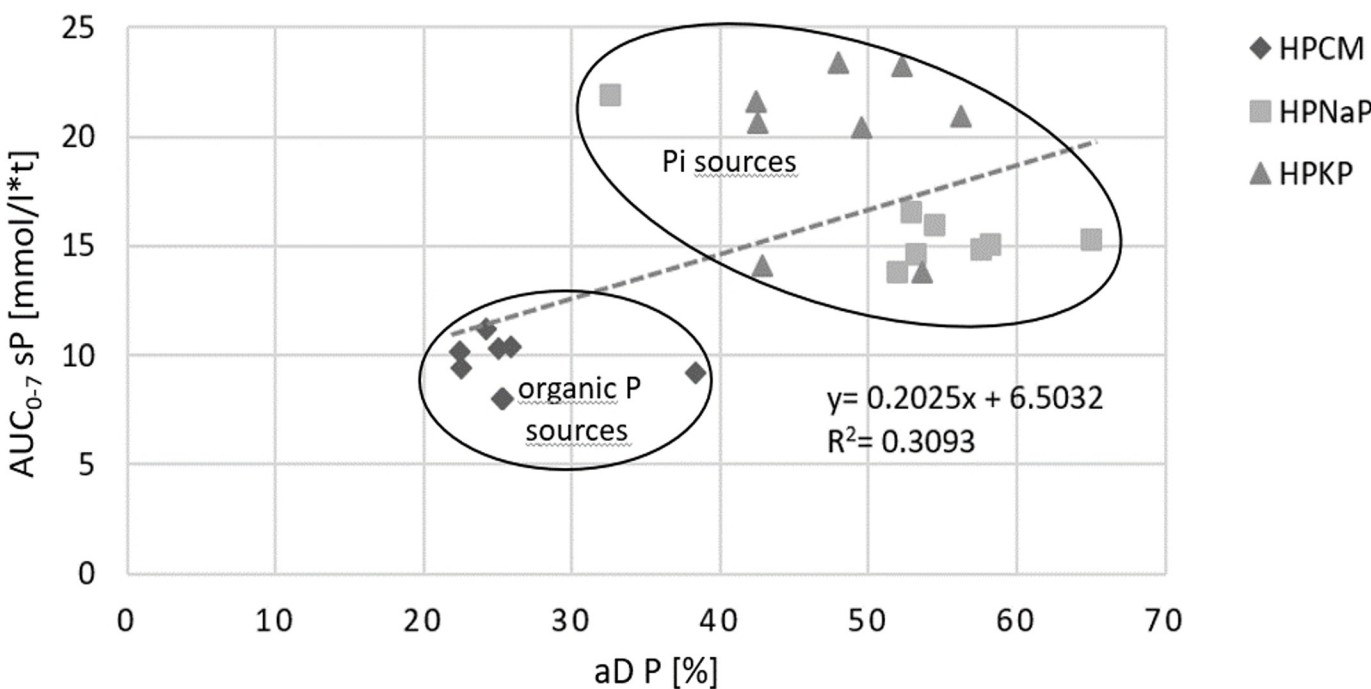

**Fig 5. Correlation between the aD P [%] and the $AUC_{0-7}$ (sP) [mmol/l*t] in dogs after feeding one control diet and 3 different HP diets containing different phosphorus sources.** aD = apparent digestibility, HP = high phosphorus diet.

people with CKD [57]. Hyperphosphatemia has several adverse effects, such as the calcification of soft tissues in kidneys and vessels. The associated higher risk of death is well documented in various species (humans: [109–111], mouse: [112], cat: [42], dog: [51,63]). The results of individuals with increased P burden through oral intake and in those with early signs of CKD are similar. Increasing sP values cause higher PTH secretion [113] in order to increase renal P excretion and to balance sCa and sP concentrations. In addition to the reduction of the sensitivity of PTH receptors (rat: [114]), high sP itself can induce compensatory hyperparathyroidism [14,115]. According to Cortadellas et al. (2010), sP values of 4.5 to 5.5 mg/dl (≙1.45–1.78 mmol/l) and higher predict the development of SRH in CKD; 84% of the dogs developed SRH under these conditions [116]. The authors therefore recommend that sP levels below 4.5 mg/dl (≙1.45 mmol/l) should be maintained in early stages of CKD to prevent SRH. In our study, values of up to 4.8 mmol/l were caused in clinical healthy dogs by adding certain Pi sources to the diet. In agreement with Siedler (2018), $KH_2PO_4$ and $NaH_2PO_4$ caused significantly higher $AUC_{0-7}$ sP [71]. The percentage of sP values above reference range was higher in diet HPNaP and HPKP than in CON and HPCM (83 and 78 vs. 24 and 29%) and, therefore, possibly harmful.

A major determinant for sCa regulation is PTH, by increasing bone resorption through stimulating osteoclast activity. In the present study, sCa was kept within tight ranges in the HPNaP and HPKP treatments, despite a significant increase of sP and PTH. In humans, hyperphosphatemia acts as a determinant for increased sCaP [117]. A sCaP of $\geq 55$ $mg^2/dl^2$ increases the risk of soft tissue calcification [55] and has therefore also been correlated to the staging system according to the International Renal Interest Society (IRIS) in dogs [116,118]. Lameness due to calcification in the paws in feline and canine CKD patients has been associated with sCaP $> 70$ $mg^2/dl^2$ [119] and shown to be a negative prognostic marker for dogs with CKD, indicating a worse outcome when $> 77$ $mg^2/dl^2$ [118]. Locatelli et al. (2002)

recommends to keep the sCaP in physiological ranges in human renal patients to prolong survival time [120]. The present study's sCaP in HPNaP and HPKP treatments exceeded all three mentioned thresholds clearly with a product of up to 180 mg$^2$/dl$^2$. Due to the short duration of the trials, no soft tissue calcifications were expected. However, it can be expected that even in healthy animals, soft tissue and vascular calcification may develop over time when such high sCaP values are caused by feeding. Especially when processed diets intended for lifelong feeding are used, even lower sCaP caused by prolonged daily periods with above normal sP, should probably be avoided. It may also be recommendable to monitor this parameter to evaluate if the dog is at risk to develop renal insufficiency.

In accordance with Siedler (2018), a significant ppr increase of PTH was detected in HPNaP and HPKP [71]. In addition, both diets caused the highest values for CrossLaps and bALP, indicating increased bone turnover. Determination of further markers of bone turnover, as recommended by Allen (2003; [121], might have been useful but not feasible due to limited available amounts of serum. Nevertheless elevated CrossLaps may indicate CKD-MBD in response to the high P load [54]. Moreover, as a potent phosphatonin, PTH accelerated renal P excretion after Pi feeding, as demonstrated by the difference between prpr and ppr P/Crea ratios. This increase in excretion was also identified by Siedler (2018; [71]). To prevent polyuria masking high P excretion, for example in diets with high Na concentrations, 24 h sampling or determination of the P/Crea ratio is recommended. To measure the complete P balance would have been preferred to be able to determine if the feeding of Pi leads to an increased P retention of the animal. However, the body has to handle the P burden caused by increased P resorption either by calcification or by renal excretion. A calcification removes soluble P from the circulating system by forming Ca-P-complexes, which includes soft tissue calcification, e.g. by osteochondrogenic differentiation of vascular smooth muscle cells [122]. A high P concentration in the urine is thought to affect also glomerular endothelial cells [123].

At high dietary P loads, FGF23 is synthesized by osteocytes, even before secretion of PTH [24,25,124]. These results are in accordance with Eller et al. (2011), who reported similar findings in a mouse model for P nephropathy [125].

The FGF23 ELISA kit used in the present study measures human intact FGF23 and has been validated for dogs [67]. Harjes et al. (2017) investigated FGF23 concentrations in dogs with CKD in different IRIS stages and in 10 healthy controls [67]. The latter had FGF23 concentrations of 315 (211–449) pg/ml, which is similar to our data for dogs in the CON treatment. Addition of Pi led to values up to 1655 pg/ml, which is otherwise found in dogs with IRIS stage 3 [67]. Interestingly, the prpr values (more than 23 h after feeding the HPi diets) were still significantly elevated in the present study. In rats, dietary induced hyperphosphatemia also caused increasing FGF23 concentrations [126]. Hardcastle and Dittmer (2015) postulate that FGF23 reacts on acute rather than on long-term hyperphosphatemia [31]. Similar to humans [30], studies in cats were not able to detect a rise in FGF23 concentrations after feeding Pi sources [49,68]. FGF23 stimulates secretion of PTH by inhibiting production of 1,25-(OH)$_2$-D$_3$ and reinforces–together with PTH–the renal P excretion. FGF23 can only bind to its receptor (FGFR) when its cofactor αklotho is present. At advanced stages of CKD, the synthesis of αklotho declines, reducing the binding capacity of FGFR and thus increasing the circulating FGF23 while hyperphosphatemia remains [127]. Moreover, downregulation of the αklotho–FGFR complex and an increase of circulating FGF23 further stimulates the secretion of PTH in rats with experimentally induced CKD [26]. This generates a vicious circle, finally causing SRH and MBD-CKD in both humans and cats [128]. Recent studies indicate that elevated FGF23 concentrations are linked to cardiovascular disease [125], and are therefore correlated with mortality in CKD patients [58,129]. The severely increased FGF23 levels seen in this study are clearly alarming as they can be interpreted as early marker for disorders of Ca

and P homeostasis in otherwise healthy dogs [67]. It seems likely that a prolonged increase of serum FGF23 and PTH levels associated with dietary Pi intake could harm dogs, and therefore cannot be declared as safe.

Reports of extreme adverse effects, including renal failure and death, in association with Pi load exist in various species (human [130,131]; mice [112,132]; rabbits [133]; dogs [51,52,102] and others). In dogs, research performed several decades ago found that oral intake of Pi can harm the kidneys [52,134]. The amount of Pi in our trials was considerably lower than in the experiments of Schneider et al. (1980; 0.21 ± 0.006 g Pi vs. 0.8 g $K_2HPO_4$/kg BW/d [52]), but effects on serum parameters were still substantial, with significant differences between Pi sources and the organic P source and CON, respectively. We hypothesize that the solubility determines to a large extend about the biological effect of the P intake on the animal, making the accompanying cation of the P salt less relevant as long as the source is highly soluble and dissolves immediately. Therefore, also other highly soluble Pi sources are expected to cause comparable effects in dogs. Furthermore, given the fact that most dogs receive multiple meals per day, the possibly threatening impact on parameters of P homeostasis might exist permanently. In accordance with Osuka and Razzaque (2012) as well as others, we hypothesize that an increase of relevant parameters of P homeostasis, especially sP, sCaP, PTH and FGF23, for an extended part of the day, might exaggerate the situation in (undiagnosed) renal patients but also affect healthy dogs. Because the dogs were not euthanized, histopathological examinations were not performed to verify this hypothesis.

## Conclusion

The results of this study support previous research on the effects of highly soluble Pi sources on the P metabolism of adult healthy dogs [71]. Moreover, the kinetics of sP, sCa, sCaP, PTH, FGF23, CrossLaps and bALP over 7 h ppr were determined in a control and in three different HP diets in dogs for the first time. The amount and source of P in the dietary treatments had a significant effect on P homeostasis, which could pose a risk for renal and skeletal health over an extended period. Thus, we presume that their use in pet food is not unconditionally safe, based on the observed disruption of P homeostasis in case of considerable amounts of ingested highly soluble Pi salts, especially when fed a lifetime. More work is needed to further evaluate consequences of acute and long-term Pi intake on pet health and, if any, no-effect-levels can be defined.

## Supporting information

**S1 Table. Serum calcium (sCa) concentrations [mmol/l] from pre- (t = 0) and up to 7 hours postprandially in adult healthy dogs fed a control (CON) and 3 high phosphorus diets, containing either poultry carcass meal (HPCM), $NaH_2PO_4$ (HPNaP) or $KH_2PO_4$ (HPKP) as a P source, for 18 days.**
(DOCX)

**S2 Table. $AUC_{0-7}$ for serum concentrations of minerals in adult dogs fed a control (CON) and 3 high phosphorus diets, containing either poultry carcass meal (HPCM), $NaH_2PO_4$ (HPNaP) or $KH_2PO_4$ (HPKP) as a P source, for 18 days.**
(DOCX)

**S3 Table. Serum parathyroid hormone (PTH) concentrations [ngl/l] from pre- (t = 0) and up to 7 hours postprandially in adult healthy dogs fed a control (CON) and 3 high phosphorus diets, containing either poultry carcass meal (HPCM), $NaH_2PO_4$ (HPNaP) or**

$KH_2PO_4$ (HPKP) as a P source, for 18 days.
(DOCX)

**S4 Table. Serum crosslaps (CL) concentrations [ng/ml] from pre- (t = 0) and up to 7 hours postprandially in adult healthy dogs fed a control (CON) and 3 high phosphorus diets, containing either poultry carcass meal (HPCM), $NaH_2PO_4$ (HPNaP) or $KH_2PO_4$ (HPKP) as a P source, for 18 days.**
(DOCX)

**S5 Table. Serum bone alkaline phosphatase (bALP) concentrations [U/l] from pre- (t = 0) and up to 7 hours postprandially in adult healthy dogs fed a control (CON) and 3 high phosphorus diets, containing either poultry carcass meal (HPCM), $NaH_2PO_4$ (HPNaP) or $KH_2PO_4$ (HPKP) as a P source, for 18 days.**
(DOCX)

**S6 Table. Serum phosphorus (sP) concentrations [mmol/l] from pre- (t = 0) and up to 7 hours postprandially in adult healthy dogs fed a control (CON) and 3 high phosphorus diets, containing either poultry carcass meal (HPCM), $NaH_2PO_4$ (HPNaP) or $KH_2PO_4$ (HPKP) as a P source, for 18 days.**
(DOCX)

**S7 Table. Serum sodium (sNa) concentrations [mmol/l] from pre- (t = 0) and up to 7 hours postprandially in adult healthy dogs fed a control (CON) and 3 high phosphorus diets, containing either poultry carcass meal (HPCM), $NaH_2PO_4$ (HPNaP) or $KH_2PO_4$ (HPKP) as a P source, for 18 days.**
(DOCX)

**S8 Table. Serum potassium (sK) concentrations [mmol/l] from pre- (t = 0) and up to 7 hours postprandially in adult healthy dogs fed a control (CON) and 3 high phosphorus diets, containing either poultry carcass meal (HPCM), $NaH_2PO_4$ (HPNaP) or $KH_2PO_4$ (HPKP) as a P source, for 18 days.**
(DOCX)

**S9 Table. Serum calcium by phosphorus product (sCaP) concentrations [$mg^2/dl^2$] from pre- (t = 0) and up to 7 hours postprandially in adult healthy dogs fed a control (CON) and 3 high phosphorus diets, containing either poultry carcass meal (HPCM), $NaH_2PO_4$ (HPNaP) or $KH_2PO_4$ (HPKP) as a P source, for 18 days.**
(DOCX)

**S10 Table. Serum fibroblast growth factor 23 (FGF23) concentrations [pg/ml] from pre- (t = 0) and up to 7 hours postprandially in adult healthy dogs fed a control (CON) and 3 high phosphorus diets, containing either poultry carcass meal (HPCM), $NaH_2PO_4$ (HPNaP) or $KH_2PO_4$ (HPKP) as a P source, for 18 days.**
(DOCX)

**S11 Table. $AUC_{0-7}$ for the serum parameters parathyroid hormone (PTH), fibroblast growth factor 23 (FGF23), bone alkaline phosphatase (bALP) and crosslaps (CL) in adult dogs fed a control (CON) and 3 high phosphorus diets, containing either poultry carcass meal (HPCM), $NaH_2PO_4$ (HPNaP) or $KH_2PO_4$ (HPKP) as a P source, for 18 days.**
(DOCX)

## Author Contributions

**Conceptualization:** Britta Dobenecker.

**Data curation:** Sven Reese.

**Investigation:** Britta Dobenecker, Sarah Herbst.

**Methodology:** Britta Dobenecker.

**Project administration:** Britta Dobenecker.

**Supervision:** Britta Dobenecker.

**Writing – original draft:** Britta Dobenecker, Sarah Herbst.

**Writing – review & editing:** Britta Dobenecker, Sarah Herbst.

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
