## [Decision Letter · Decision Letter 0]

12 Jan 2021

PONE-D-20-35745

Effects of dietary phosphates from organic and inorganic sources on parameters of phosphorus homeostasis in healthy adult dogs

PLOS ONE

Dear Dr. Dobenecker,

Thank you for submitting your manuscript to PLOS ONE. After careful consideration, we feel that it has merit but does not fully meet PLOS ONE’s publication criteria as it currently stands. Therefore, we invite you to submit a revised version of the manuscript that addresses the points raised during the review process.

Please address the comments raised by the 2nd reviewer.

We look forward to receiving your revised manuscript.

Kind regards,

Ahmad N. Al-Dissi, BVetSc, MSc, PhD, DACVP

Academic Editor

PLOS ONE

Journal Requirements:

2. In your Methods section, please include a comment about the state of the animals following this research. Were they housed for use in further research?

Additional Editor Comments:

Thanks for the chance to examine this work. Please address the comments of the 2nd reviewer.

Reviewers' comments:

Reviewer's Responses to Questions

**Comments to the Author**

1. Is the manuscript technically sound, and do the data support the conclusions?

Reviewer #1: Yes

Reviewer #2: Yes

2. Has the statistical analysis been performed appropriately and rigorously? 

Reviewer #1: Yes

Reviewer #2: Yes

3. Have the authors made all data underlying the findings in their manuscript fully available?

Reviewer #1: Yes

Reviewer #2: Yes

4. Is the manuscript presented in an intelligible fashion and written in standard English?

Reviewer #1: Yes

Reviewer #2: Yes

5. Review Comments to the Author

Reviewer #1: The number of "test" dogs in this study is low, however you were able to achieve statistically significant data given the disparity of parameters in the dogs receiving inorganic phosphorus supplementation. I think a larger study with more dogs would provide improved data overall, however these initial findings should be reported in the literature and would be of interest to readers.

This is a well-designed study with good execution and interesting results.

I had several questions that arose while reading the manuscript and before I had a chance to pose them to you in the review, I found the answer in the discussion.

The discussion piece surrounding the potential relationship with Pi and the sCAP product was very interesting.

Solid conclusions based on the available results.

Reviewer #2: The authors investigated the potential impact of high phosphorus diets (organic versus inorganic) on Ca/P homeostasis in a small group of dogs. This is particularly relevant given that iP may be added to commercial diets. As such, this study makes a relevant and important contribution to the literature. Overall, the study is well thought out, thorough, and detailed. There are only a few minor issues.

line 162 - the authors describe how Ca, Na, and K were measured, but how was serum P measured? Could you also elaborate on blood sample handling procedures? i.e. how quickly was serum removed from cells? Were the samples screened for hemolysis (to rule out preanalytical artefact)

line 337 - In the legend for Fig 5, it may be helpful to define the acronyms for aD and HP for the reader

line 392 - the sentence beginning with "Nevertheless..." is confusing, is the word "may" misplaced?

line 443 - would improve readability by replacing "sol" with soluble

6. PLOS authors have the option to publish the peer review history of their article (what does this mean?). If published, this will include your full peer review and any attached files.

Reviewer #1: No

Reviewer #2: No

---

## [Author Response · Author response to Decision Letter 0]

15 Jan 2021

We want to thank the editor as well as both reviewers for the interest, the positive feedback as well as the encouraging words!

The format was amended according to the Journal’s requirements.

Further comments Reviewer 2:

line 162 - the authors describe how Ca, Na, and K were measured, but how was serum P measured? 

We indeed missed to name serum here. Now added

Could you also elaborate on blood sample handling procedures? i.e. how quickly was serum removed from cells? Were the samples screened for hemolysis (to rule out preanalytical artefact) 

The following sentences were added: Blood samples were allowed to clot for 20 minutes at room temperature before centrifugation (1.260 g) for 10 minutes and serum was aliquoted into several Eppendorf safe-lock cups. For FGF23 measurements, whole blood was allowed to clot for 2 hours before centrifugation at 560 g for 15 minutes. Preanalytical artefacts due to haemolytic samples can be ruled out.

line 337 - In the legend for Fig 5, it may be helpful to define the acronyms for aD and HP for the reader 

Thank you for the comment. The explanation for the abbreviations were added

line 392 - the sentence beginning with "Nevertheless..." is confusing, is the word "may" misplaced? 

You are right. The sentence was corrected

line 443 - would improve readability by replacing "sol" with soluble 

Thank you! Corrected

---

## [Decision Letter · Decision Letter 1]

29 Jan 2021

Effects of dietary phosphates from organic and inorganic sources on parameters of phosphorus homeostasis in healthy adult dogs

PONE-D-20-35745R1

Dear Dr. Dobenecker,

We’re pleased to inform you that your manuscript has been judged scientifically suitable for publication and will be formally accepted for publication once it meets all outstanding technical requirements.

Kind regards,

Ahmad N. Al-Dissi, BVetSc, MSc, PhD, DACVP

Academic Editor

PLOS ONE

Additional Editor Comments (optional):

Reviewers' comments:

Reviewer's Responses to Questions

**Comments to the Author**

1. If the authors have adequately addressed your comments raised in a previous round of review and you feel that this manuscript is now acceptable for publication, you may indicate that here to bypass the “Comments to the Author” section, enter your conflict of interest statement in the “Confidential to Editor” section, and submit your "Accept" recommendation.

Reviewer #2: All comments have been addressed

2. Is the manuscript technically sound, and do the data support the conclusions?

Reviewer #2: Yes

3. Has the statistical analysis been performed appropriately and rigorously? 

Reviewer #2: Yes

4. Have the authors made all data underlying the findings in their manuscript fully available?

Reviewer #2: Yes

5. Is the manuscript presented in an intelligible fashion and written in standard English?

Reviewer #2: Yes

6. Review Comments to the Author

Reviewer #2: (No Response)

7. PLOS authors have the option to publish the peer review history of their article (what does this mean?). If published, this will include your full peer review and any attached files.

Reviewer #2: No

---

## [Editor Report · Acceptance letter]

3 Feb 2021

PONE-D-20-35745R1 

Effects of dietary phosphates from organic and inorganic sources on parameters of phosphorus homeostasis in healthy adult dogs 

Dear Dr. Dobenecker:

I'm pleased to inform you that your manuscript has been deemed suitable for publication in PLOS ONE. Congratulations! Your manuscript is now with our production department. 

Kind regards, 

on behalf of

Dr. Ahmad N. Al-Dissi 

Academic Editor

PLOS ONE